# Tuning crystal-phase of bimetallic single-nanoparticle for catalytic hydrogenation

Shuang Liu [1], Yong Li [1] ✉, Xiaojuan Yu[2], Shaobo Han[1], Yan Zhou [1], Yuqi Yang[3], Hao Zhang[3], Zheng Jiang [3] ✉, Chuwei Zhu[4], Wei-Xue Li [4], Christof Wöll [2], Yuemin Wang [2] ✉ & Wenjie Shen [1] ✉

Bimetallic nanoparticles afford geometric variation and electron redistribution via strong metal-metal interactions that substantially promote the activity and selectivity in catalysis. Quantitatively describing the atomic configuration of the catalytically active sites, however, is experimentally challenged by the averaging ensemble effect that is caused by the interplay between particle size and crystal-phase at elevated temperatures and under reactive gases. Here, we report that the intrinsic activity of the body-centered cubic PdCu nanoparticle, for acetylene hydrogenation, is one order of magnitude greater than that of the face-centered cubic one. This finding is based on precisely identifying the atomic structures of the active sites over the same-sized but crystal-phase-varied single-particles. The densely-populated Pd-Cu bond on the chemically ordered nanoparticle possesses isolated Pd site with a lower coordination number and a high-lying valence $d$-band center, and thus greatly expedites the dissociation of $H_2$ over Pd atom and efficiently accommodates the activated H atoms on the particle top/subsurfaces.

Bimetallic catalysts, consisting of a platinum-group metal and a late-transition metal, featured geometrical variation and electronic redistribution via metal–metal interactions and enhanced the activity and/or selectivity, strongly depending on the crystal-phase and the particle size[1–9]. PdCu nanoparticles in the chemically ordered body-centered cubic (B2) phase, for instance, showed pronouncedly increased activity (2–20 times) for the electro-conversion of energy-related molecules[10–15] and substantially promoted selectivity for the hydrogenation of multiple carbon–carbon bonds[16], relative to the disordered face-centered cubic (fcc) ones. Experimental studies and theoretical calculations on PdCu model systems, mostly large, single-crystals or extended surfaces, have presumably proposed that Cu atom modulated the electron density and the geometric arrangement of Pd atom and thus altered the activation pathways of the reacting molecules[17–21]. Investigations on the mechanism of Pd-Cu interaction in powder PdCu catalysts were commonly done by altering the chemical composition[11,22–26] and/or the particle size[10,12–16,27–32]. These alloyed particles typically adopted the disordered fcc phase that allowed inherently randomly mixing Pd and Cu atoms; their transformation to the chemically ordered B2 phase was achieved by heating to high temperatures under inert/reducing gases. However, the interplay between particle size and crystal-phase caused significant diffusion of Pd and Cu atoms among the particles (particle columniations), and thus yielded a dual-size distribution, with B2 particles having lager while fcc particles having smaller sizes. To date, elucidating the precise structure of the active sites over bimetallic nanoparticles presents a huge experimental challenge because of the particle communications during preparation and catalysis.

In this work, we quantified the geometric structure and the electronic character of the active sites over same-sized but crystal-phase-

[1]State Key Laboratory of Catalysis, Dalian Institute of Chemical Physics, Chinese Academy of Sciences, Dalian, China. [2]Institute of Functional Interfaces, Karlsruhe Institute of Technology, Eggenstein-Leopoldshafen, Germany. [3]Shanghai Synchrotron Radiation Facility, Shanghai Advanced Research Institute, Chinese Academy of Sciences, Shanghai, China. [4]School of Chemistry and Materials Science, Hefei National Research Center for Physical Sciences at the Microscale, University of Science and Technology of China, Hefei, China. ✉e-mail: yongli@dicp.ac.cn; jiangzheng@sinap.ac.cn; yuemin.wang@kit.edu; shen98@dicp.ac.cn

varied PdCu single-particles, using a complementary combination of aberration-corrected scanning transmission electron microscopy (STEM), X-ray absorption near-edge structure (XANES), extended X-ray absorption fine structure (EXAFS) spectroscopy, infrared (IR) spectroscopy and density functional theory (DFT) calculations. We have found that the ordered B2 particle (8 nm) promoted the intrinsic activity, for acetylene hydrogenation, by one order of magnitude, as compared to the disordered fcc particle. The densely-populated surface Pd-Cu bond, defined by the chemically ordered crystal-phase, possessed isolated Pd site with a lower coordination number and a high-lying metal $d$-band center. It not only greatly expedited $H_2$ dissociation on the Pd atom but also effectively accommodated the activated H atoms on the particle top/subsurfaces, as evidenced by the direct observation of H atoms in environmental TEM. This provides a new approach to study single-nanoparticle catalysis with a chemical reaction by precisely accounting for the atomic configuration of active site on the particle surface.

## Results and discussion

### Tuning crystal-phase of single-nanoparticles

We tuned the crystal-phase of PdCu catalysts at the single-nanoparticle scale that allowed only intraparticle atomic rearrangement (Fig. 1). Monodisperse PdCu colloids of around 8.0 nm in the B2 phase were initially crystallized in ethylene glycol (Supplementary Figs. 1 and 2); each colloid was then precisely coated with a silica shell via a reverse microemulsion method. $H_2$ treatment of this core-shell structured precursor at 673 K yielded the B2 particle that consisted of a metal core of 8.6 nm in diameter and a silica shell of 7.1 nm thick (Supplementary Fig. 3). The fcc particle was obtained by treating the B2 particle with $O_2$ at 673 K (Supplementary Fig. 4), followed by $H_2$ reduction at 773 K, resulting in a metal core of 8.1 nm and a silica shell of 7.7 nm (Supplementary Fig. 5). The permeable porous silica shell, formed during the high-temperature treatments with the reactive gases ($H_2$/$O_2$), spatially confined the metal particle and thus persevered the particle size, but still allowed diffusion of

small molecules to the metal surface for studying their chemical transformation.

Aberration-corrected high-angle annular dark field STEM (HAADF-STEM) analysis verified the spherical shape of the B2 particle. The lattice spaces of 0.21 and 0.29 nm with a dihedral angle of 45° (Fig. 2a), viewed along the [001] direction, referred to the {110} and {100} facets, respectively. As viewed along the [11$\bar{1}$] direction, the two lattice spaces of 0.21 nm, with a dihedral angle of 120° (Fig. 2b), indicated the {110} facets. Accordingly, the B2 particle was projected to be enclosed by twelve {110} facets with an ordered arrangement of Pd and Cu atoms and six {100} facets terminated by either Pd or Cu atoms (Fig. 2c). Detailed STEM analysis on the fcc particle identified that it was terminated by eight {111} facets and six {100} planes (Fig. 2d–f). The dominantly exposed {111} facets were characterized by the two lattice spacings of 0.21 nm with a dihedral angle of 110° while the minor {100} facets were indexed by the lattice spacings of 0.19 nm.

### Coordination environments of Pd and Cu atoms

XANES and EXAFS have revealed the bond distances and the electronic features of the PdCu particles. The XANES spectra of Pd and Cu K-edges in the B2 and fcc particles indicated the metallic nature by sharing similar absorption patterns to Cu or Pd foils; simulations on the experimental data assured that the arrangements of Pd and Cu atoms followed the respective crystal-phases (Supplementary Fig. 6). A quantitative analysis on the EXAFS spectra at Cu and Pd K-edges found that the coordination environments of the metal atoms were fully consistent with the atomic structures defined by the crystal-phases (Fig. 3a, b). For the B2 particle, the first nearest neighbor (1 NN) Pd−Cu bond length amounted to 2.56 Å, with a coordination number (CN) of 6.1, signifying that Pd atoms were coordinated with only Cu atoms and vice versa[11,25,33]. The Pd-Pd bond length of 2.98 Å and the Cu−Cu bond length of 2.92 Å in the second nearest neighbor (2NN), with CNs of 5.5 and 6.0, respectively, were considerably greater than those in the monometallic Pd (2.74 Å) and Cu (2.54 Å) particles[34,35]. This means that Pd (Cu) is atomically dispersed in Cu (Pd) lattice and forms ordered

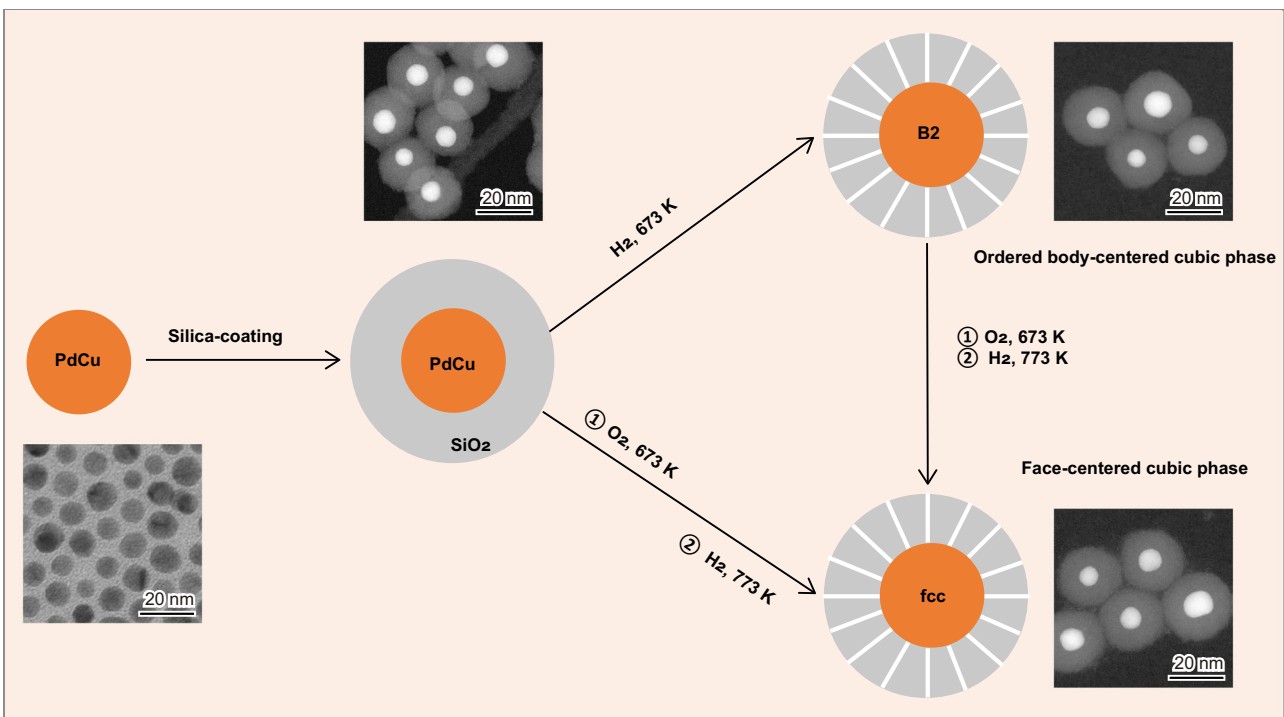

**Fig. 1 | Tuning crystal-phase of the PdCu catalysts at the single-nanoparticle scale.** The starting PdCu colloid (8.0 nm) had a Pd/Cu molar ratio of 1/1 and an ordered body-centered cubic (B2) phase. The TEM images illustrate the representative structures for the respective samples.

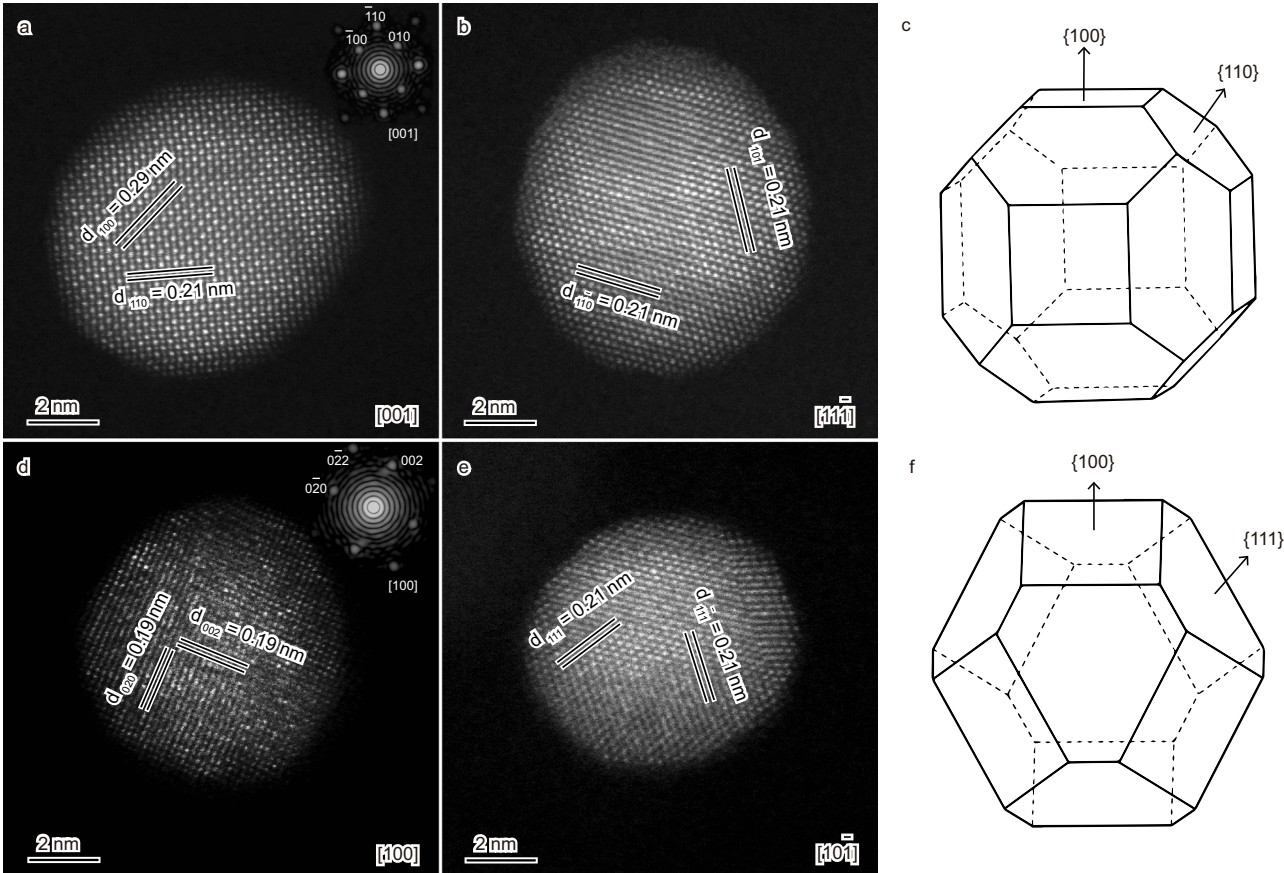

**Fig. 2 | Geometric structures of the PdCu nanoparticles. a–c** STEM images of the B2 particle viewed along the [001] (**a**) and [11$\bar{1}$] (**b**) directions and the projected morphology (**c**) that is enclosed by twelve {110} facets and six {100} facets. **d–f** STEM images of the fcc particle viewed along the [100] (**d**) and [10$\bar{1}$] (**e**)

orientations and the projected morphology (**f**) terminated by eight {111} facets and six {100} planes. The insets in (**a**) and (**d**) are the corresponding fast Fourier transform (FFT) patterns that showed the crystalline features of the respective particles.

Pd–Cu bond over the B2 particle. In sharp contrast to this almost perfect ordering, the fcc particle had a random mixture of Pd and Cu atoms. The 1NN Pd–Cu bond had a length of 2.61 Å with a CN of 4.5; the 1NN Cu–Cu bond (2.60 Å) had a CN of 5.7 and the 1NN Pd–Pd bond (2.66 Å) had a CN of 5.3. These results suggest that the shell of nearest neighbor of Pd consists of both Cu and Pd atoms. In other words, Cu (Pd) is simultaneously coordinated by Cu and Pd, yielding a co-existence of Pd–Cu, Pd–Pd and Cu–Cu bonds with a total CN of around 10[33].

Our DFT calculations found the total density of states (DOS) of the B2 bulk upshifted towards the Fermi level, as compared to the fcc bulk (Fig. 3c and Supplementary Fig. 7). Specifically, Pd and Cu atoms in the B2 phase had substantially higher energies for the valence *d*-band centers (−2.08 and −1.79 eV), relative to the fcc phase (−2.34 and −2.07 eV). These upshifts stemmed from the lower coordination numbers in the 1 NN, 8 for the B2 phase vs 12 for the fcc phase. A similar trend was found by the EXAFS data for the corresponding PdCu nanoparticles. In the B2 phase Pd–Cu bond is prevailed exclusively in the 1NN, whereas the fcc phase is constructed by half of Pd–Cu bond and another half of either Pd–Pd or Cu–Cu bonds in the 1 NN, leading to a totally different orbital hybridization between the two metals. The difference was further observed in the exposed surface metal atoms (Supplementary Fig. 8). For the surface Pd and Cu atoms on B2(110), the corresponding valance *d*-bands upshifted further to −1.74 and −1.59 eV, respectively, and remained higher than those on fcc(111), −1.96 and −1.84 eV. A metal surface with a high lying valence *d*-band center is well documented to be more active[36–38]. To provide more insights into Pd-Cu interaction, integral of crystal orbital Hamiltonian

population (ICOHP) below the Fermi level was used to quantify their bonding strength (Fig. 3d). Calculated absolute ICOHP of 0.55 eV for surface Pd–Cu bond on B2(110) was smaller than that of fcc(111) (1.00 eV). This tells that the corresponding Pd–Cu bond on B2(110) is relative weak, although the bonding strengths of Pd with subsurface Cu are essentially same for B2(110) and fcc(111). As a compensation, a higher surface reactivity for B2(110) is expected, in line with analysis from the *d*-band center.

## Geometric configurations and electronic interactions of surface Pd and Cu atoms

The geometric and electronic properties of Pd and Cu atoms on the outermost layers of the particles were examined by IR spectroscopy using CO as the probe molecule. Upon exposure of the B2 particle to CO at 180 K, distinct CO bands on Cu (2122/2105 cm$^{-1}$) and Pd (2063/1915 cm$^{-1}$) were observed (Fig. 4a). For the Cu-related vibrations, the band at 2122 cm$^{-1}$ referred to CO on isolated Cu site surrounded by Pd atoms; while the signal at 2105 cm$^{-1}$ was ascribed to CO bound at Cu ensemble, like a contiguous monolayer[24,39,40]. Over Pd, the intense band at 2063 cm$^{-1}$ represented CO bound in an on-top fashion to isolated Pd site that coordinated to Cu atoms; the band at 1915 cm$^{-1}$ indicated bridge-bonded CO on contiguous Pd atoms[24,40–42]. These spectroscopic results, along with the STEM/EXAFS data, reaffirmed that the dominant {110} facet on the B2 particle was terminated by isolated Pd and Cu atoms while the minor {100} facet was enriched with either Cu or Pd monolayers. The IR spectrum of CO adsorption on the fcc particle at 180 K differed substantially from that for the B2 particle, illustrating the significantly changed coordination

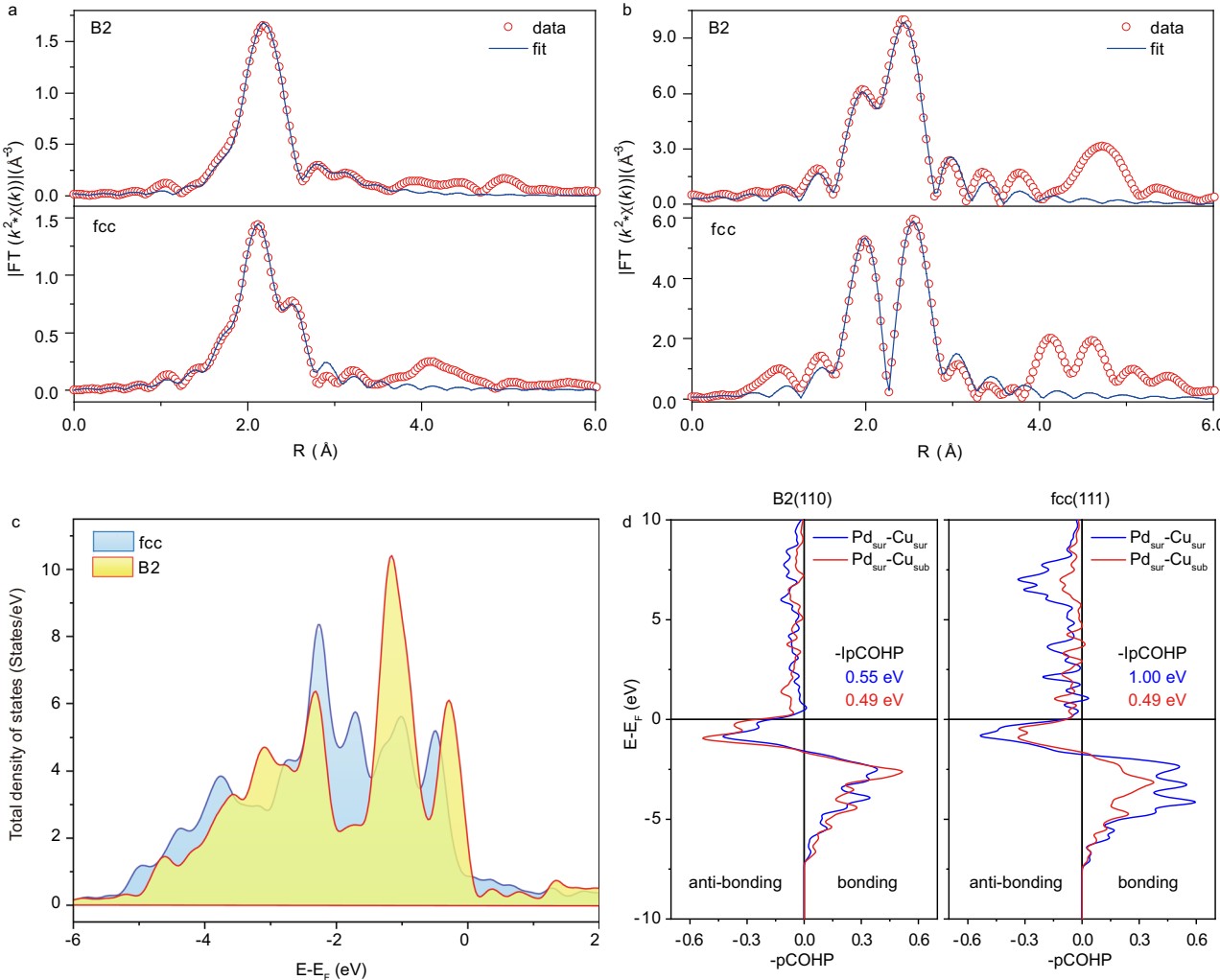

**Fig. 3 | Bonding environments of Pd and Cu atoms in the PdCu nanoparticles.** **a**, **b** $k^2$-weighted Fourier transformed EXAFS spectra of Pd (**a**) and Cu (**b**) K-edges; the red circles show the experimental data while the blue curves indicate the fitting results. **c** Total density of states per PdCu formula, showing the upshift of the B2 bulk with respect to the fcc bulk due to the lower coordination numbers in the first nearest neighbor. **d** Projected crystal orbital Hamiltonian populations (COHP) between the surface Pd atoms and surface or subsurface Cu atoms of the B2(110) and fcc(111) surfaces.

environments of Cu and Pd atoms (Fig. 4b). In addition to CO adsorptions on isolated Cu site and ensemble, the new, intense band at 2091 cm$^{-1}$, typical for CO bound to Cu (111) surfaces, indicated Cu-enriched sites (multilayered aggregates)[39,43]. The band of CO bound at isolated Pd site blue-shifted to 2070 cm$^{-1}$, meaning the weakened electronic interaction with the adjunct Cu atom. Moreover, the low-frequency vibrations of CO on contiguous Pd atoms splitted into two bands at 1952 cm$^{-1}$ (bridged sites) and 1890 cm$^{-1}$ (3-fold hollow sites), showing the diversity of Pd aggregates due to the random arrangements of Cu and Pd atoms[42].

The temperature-dependent IR data shed insights into the electronic interactions of surface Cu and Pd atoms on the B2 and fcc particles and provided further evidence for the assignment of the IR bands (Supplementary Fig. 9). The weakly adsorbed CO species on Cu-related sites were less stable and completely desorbed at 260–280 K. A quantitative analysis on the on-top CO-Pd band attested the pronounced Pd-Cu electronic interaction on the B2 particle (Fig. 4c). IR spectra recorded at 300 K, where the weakly bound CO on Cu sites fully desorbed, showed a single CO-Pd band at 2044 cm$^{-1}$ for the B2 particle (Fig. 4d), which was lower in frequency by 21 cm$^{-1}$ relative to that on the fcc particle (Fig. 4e). This verified the different chemical environment of surface Pd atoms on the B2 particle. At the extremely low CO coverage at 460 K, where the coverage-induced frequency shift

was safely excluded, the on-top CO-Pd band appeared at 2028 cm$^{-1}$ on the B2 particle while at 2048 cm$^{-1}$ on the fcc particle (Supplementary Fig. 9). Both vibrations red-shifted considerably with respect to that for CO on Pd particles (2100–2050 cm$^{-1}$)[42], and outweighed any frequency shift recorded for CO adsorption at isolated Pd site (2070–2050 cm$^{-1}$) over Pd$_1$Cu single-atom alloys[24,40]. Of note, the rather low frequency of the on-top CO-Pd band (2028 cm$^{-1}$) over the B2 particle implicated a dramatically enhanced electron back-donation from the Pd-4$d$ band to the CO 2$\pi^*$ orbital, in line with the upshifted $d$-band center of Pd atoms (Fig. 3c and Supplementary Fig. 7).

## Atomic structures of the active sites for acetylene hydrogenation

These same-sized but crystal-phase-varied PdCu particles were then tested for the selective hydrogenation of acetylene to ethylene, an industrial process for eliminating acetylene in the ethylene-rich stream used for polyethylene synthesis[44–46]. For a stoichiometric H$_2$/C$_2$H$_2$ ratio of 1/1 in the feed gas, the conversion of C$_2$H$_2$ on the B2 particle was 35% at 298 K while it was only 5% on the fcc particle (Fig. 5a). As increasing the H$_2$/C$_2$H$_2$ ratio to 2/1, the conversion of acetylene on the B2 particle jumped to 92% but it just increased to 14% over the fcc particle. The selectivities of ethylene (70-80%) in both cases were practically identical (Supplementary Fig. 10), showing a typical character for Pd-based

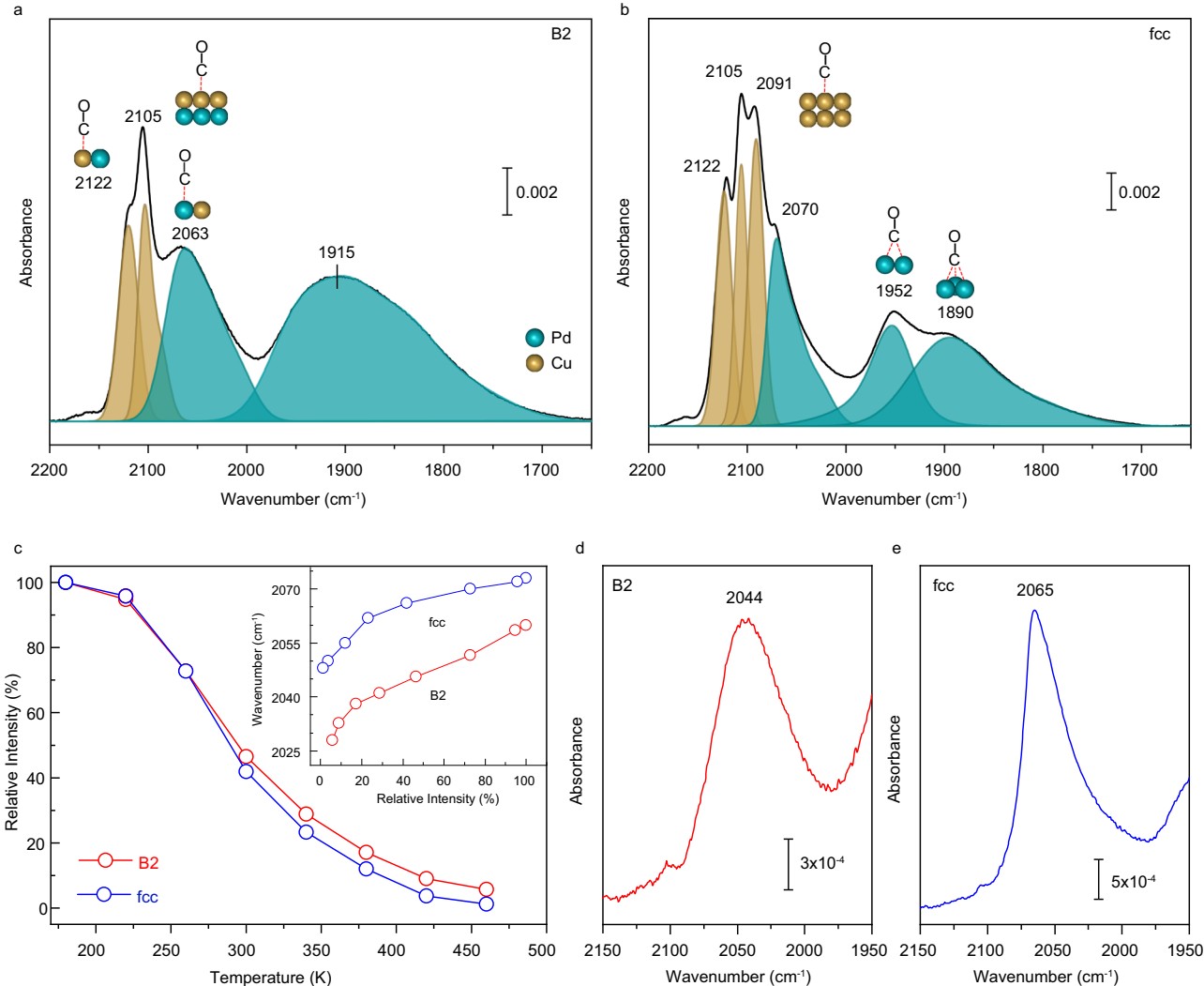

**Fig. 4 | Geometric configurations and electronic interactions of surface Pd and Cu atoms on the PdCu particles. a**, **b** IR spectra of CO adsorptions on the B2 (**a**) and fcc (**b**) particles, recorded after exposing the samples to CO at 180 K. **c** Intensity evolution of the on-top CO-Pd band, as a function of temperature. The spectra are acquired after CO adsorption on the sample at 180 K and raising the temperature to 460 K at a rate of 3 K min$^{-1}$; the insert shows that the gradual attenuation of the on-top CO-Pd band is accompanied by a distinct red-shift in frequency (32 cm$^{-1}$ for the B2 particle and 25 cm$^{-1}$ for the fcc particle at 460 K). **d**, **e** IR spectra of the on-top CO-Pd bands over the B2 (**d**) and fcc (**e**) particles at 300 K, where CO on Cu-related sites is fully desorbed.

catalysts[22,26]. The outstanding performance of the B2 particle was further demonstrated by varying the feed gas composition (Supplementary Fig. 11). The activity of the B2 particle, measured at room temperature and under a differential reactor condition, was one order of magnitude greater than that of the fcc particle as ranging the H$_2$/C$_2$H$_2$ ratio from 1/2 to 2/1 (Fig. 5b), evidencing that the B2 particle is intrinsically more active than the fcc particle. Over both particles, the activation energies varied only marginally, 44–50 kJ mol$^{-1}$, while the reaction orders were nearly unity regarding H$_2$ and slightly negative for C$_2$H$_2$ (Supplementary Fig. 12), confirming the identical reaction pathways and the rate-limiting step of H$_2$ activation[26,45,46].

Microcalorimetric adsorption experiments revealed that both PdCu particles had a saturated amount of C$_2$H$_2$ adsorbed (Supplementary Fig. 13), suggesting a similar pattern of acetylene adsorption; the abrupt change in the differential heat, as increasing C$_2$H$_2$ coverage, indicating that the adsorption geometry of C$_2$H$_2$ shifted from di-$\sigma$-bonded molecule to $\pi$-bonded state. The amount of adsorbed H$_2$, at a monolayer level, on the B2 particle was about twice of that on the fcc particle. This superior ability of the B2 particle for dissociating H$_2$ was visualized directly by aberration-corrected environmental TEM. As exposing a B2 particle to H$_2$ at 303 K, dissociated H atoms were

explicitly observed to occupy the interstitial sites between Pd and Cu atomic columns (Fig. 5c). Simulated TEM images found that there presented at least two layers of H atoms on the {110} facets, approximately equaling to a density of 33 H nm$^{-2}$ (Supplementary Fig. 14). As exposing the B2 particle to a H$_2$/C$_2$H$_2$ mixture, i.e., the reaction condition, the dissociated H atoms reacted with acetylene rapidly and the H atomic columns became invisible (Fig. 5d and Supplementary Fig. 15). On the fcc particle, H$_2$ dissociation might occur similarly at Pd atoms, but the activated H atoms were rarely observed, probably because of the lower density (Supplementary Fig. 16).

We then performed DFT calculations for H$_2$ activation on the PdCu particles. As the dominantly exposed (110) facet over the B2 particle and the (111) facet on the fcc particle have been experimentally verified to be largely responsible for acetylene hydrogenation[47,48] whereas the minor (100) facet, terminated by either Pd or Cu in both cases, was much less selective or inactive at room temperature[48,49], we adopted B2(110) and fcc(111) as the representative models (Supplementary Fig. 17). The B2(110) facet was found not only to dissociate H$_2$ actively but also promote the accumulation of the dissociated H atoms. Specifically, the adsorption energies of H$_2$ on the favorable surface Pd sites of B2(110) and fcc(111) were −0.26 and −0.10 eV,

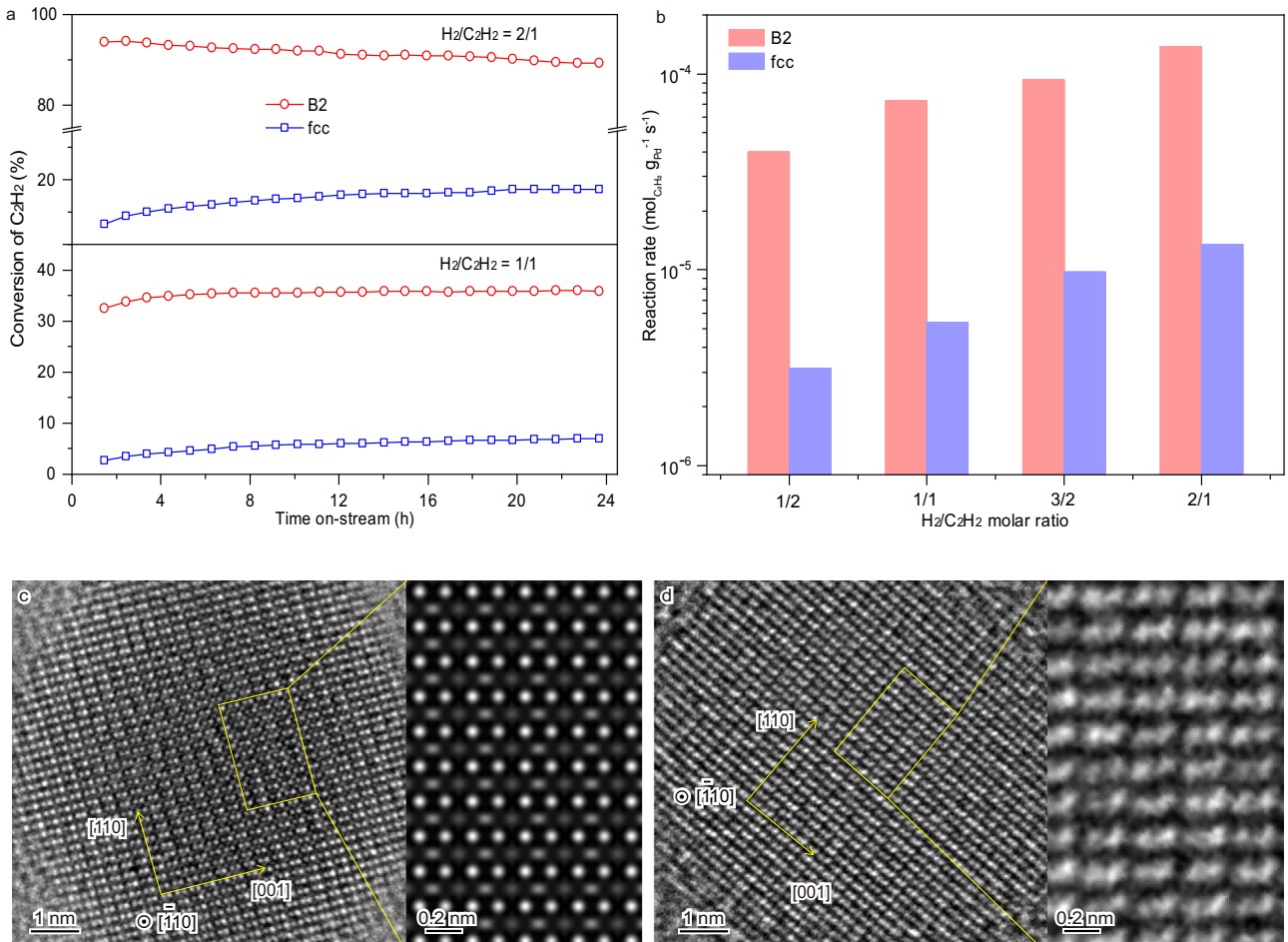

**Fig. 5 | Acetylene hydrogenation over the PdCu particles. a** Conversion of acetylene at the $H_2/C_2H_2$ molar ratios of 1/1 and 2/1 at 298 K. **b** Reaction rate as ranging the $H_2/C_2H_2$ ratio from 1/2 to 2/1 at 298 K. **c** A TEM image of a B2 particle under $H_2$ (1 mbar) at 303 K; the simulated image of the rectangular region shows the dissociated H atoms (grey dots) sited between Pd and Cu atomic columns (bright dots). **d** A TEM image of a B2 particle during acetylene hydrogenation ($H_2/C_2H_2$, 1 mbar) at 303 K; the enlarged image of the rectangular region illustrates the dynamic variations of Pd and Cu atoms but rarely hydrogen atoms. All the environmental TEM images are taken along the [$\bar{1}$10] zone axis of the B2 particle.

respectively; while the corresponding adsorption energies for the dissociated H atoms, located between Pd and Cu atoms, were −0.53 and −0.38 eV (Fig. 6a). This suggests that both molecular and dissociative adsorptions for hydrogen on B2(110) are thermodynamically more favorable than these over fcc(111). Stronger molecular adsorption on B2(110) implied a large pre-activation to $H_2$, as seen from the longer H−H bond distance (0.851 Å vs 0.833 Å, Fig. 6b, c). A larger weakening on the H−H bond was further seen from the smaller −IpCOHP (4.83 eV vs 5.09 eV, Fig. 6c). The dissociation barrier was, however, negligible for both surfaces (0.02 eV and 0.04 eV, Fig. 6b). Meanwhile, to approach the transition state, a less stretching in H−H bond was required for B2(110) (1.146 Å vs 1.254 Å) along with a larger -IpCOHP (2.34 eV vs 1.83 eV), indicating a facile kinetics as well.

Taking these structural characters and reaction performances into account, the much higher activity of the B2 particle could be ascribed to the orderly arranged Pd−Cu bond with low coordination numbers and high-lying *d*-band centers. With this regard, both experimental studies[5,19,23,24,26,50] and theoretical calculations[20,51–56] have demonstrated that the atomically-dispersed Pd atom on Cu matrices (fcc phase) enabled both facile dissociation of $H_2$ at Pd and weaker binding of dissociated H atoms on Cu, and thus improved the selectivity. These single-atom alloys were prepared by doping 0.01 monolayer Pd on Cu(111) extended surfaces[19] or small amount (0.02−2.0 mol.%) of Pd on Cu particles (2−40 nm)[23,24,26]; they featured high selectivity towards ethylene (67−95%), but required unfortunately to operate at temperatures above 323 K and over-stoichiometric $H_2/C_2H_2$ ratios (>2/1). Doping with a higher amount of Pd (10 mol.%) enhanced the activity but dramatically lowered the selectivity towards ethylene (15%) because of the aggregation of Pd atoms[5,24]. PdCu bulk alloys with a majority of Pd (50−90 mol.%), typically adopting the fcc phase, promoted the selectivity towards ethylene but at the expense of activity[46,57]. Pd atoms served as the active sites for dissociating $H_2$ while Cu atoms simply diluted Pd ensembles but did not interact electronically with Pd. Here, the B2 particle is characterized by the densely-populated surface Pd−Cu bond that possessed isolated Pd site with a lower coordination number, and thereby promoted the activity pronouncedly at room temperature under a stoichiometric feed gas condition. To some extents, it circumvented the problems encountered by the newly emerged single-atom alloys and the traditional bulk alloys for the selective hydrogenation of multiple carbon−carbon bonds.

In summary, the comprehensive experimental data and theoretical calculations justified that tuning the crystal-phase of bimetallic catalysts at the single-nanoparticle scale changed the atomic structure of the active sites. The densely-populated Pd−Cu bond on the chemically ordered B2 particle, featured by the low coordination numbers and the high-lying metal *d*-band centers, greatly facilitated $H_2$ dissociation on the Pd atom and efficiently accommodated the activated H atoms on the top/subsurfaces, yielding a much higher activity. This

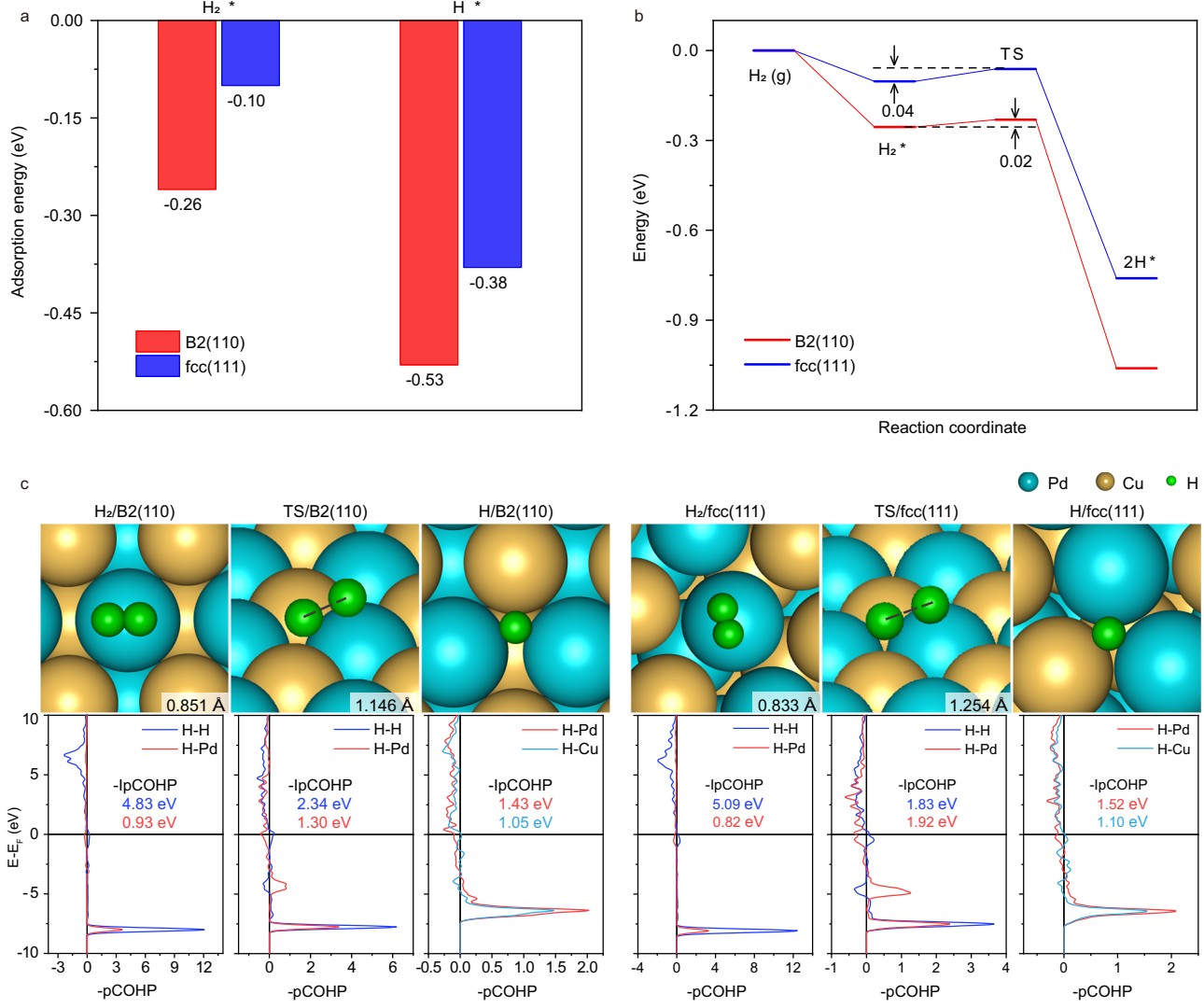

**Fig. 6 | H₂ dissociation on the B2(110) and fcc(111) surfaces. a** Adsorption energies of H₂ and H atom; **b** Potential-energy diagrams of H₂ dissociation; **c** Geometric configurations of H₂ and the dissociated H atom and the corresponding COHP curves (bottom).

finding fundamentally offers a new route to precisely tailor the geometric and electronic structures of the active sites over bimetallic particles for quantifying the structure–reactivity relationship at atomic accuracy, and practically provides an accessible approach to essentially promote the efficiency of metal catalysts by tuning their intrinsic activity via crystal-phase control at single-nanoparticle level.

## Methods

### Crystal-phase mediation of PdCu particles

Monodisperse PdCu colloids were prepared by reducing palladium and copper cations with ethylene glycol and using oleylamine as the capping agent. 66.2 mg $Na_2PdCl_4$, 46.9 mg $CuCl_2\cdot2H_2O$ and 1 ml oleylamine were dissolved into 100 ml ethylene glycol at room temperature under Ar. The mixture was heated to 393 K under stirring and maintained at that temperature for 20 min; further heated to 473 K and kept at that temperature for 2 h. The PdCu colloids were dispersed into cyclohexane (250 ml).

Each PdCu colloid was then precisely coated with a thin silica shell using a water-cyclohexane reverse microemulsion method. PdCu colloids (37.4 mg), dispersed in cyclohexane (250 ml), were mixed with Triton X-100 (polyethylene glycol tert-octylphenyl ether, 80 ml) and ultra-sonically treated for 0.5 h at 293 K. Aqueous ammonia solution (29.4 wt.%, 5 ml) and aqueous hydrazine hydrate solution (80 wt.%,

5 ml) were added to mediate the pH value of the mixture to 13. 9.35 g tetraethyl orthosilicate, mixed with 250 ml cyclohexane, was added and the suspension was stirred for 1 h. The solid product was collected by centrifugation, washed with ethanol, and dried at 323 K under vacuum for 12 h, yielding silica-coated PdCu colloid.

Crystal-phase tuning was done by treating the silica-coated PdCu colloid with reactive gases ($H_2/O_2$) at 673–773 K. The ordered body-centered cubic (B2) phase, donated as B2 particle, was prepared by treating the silica-coated PdCu colloid with $H_2$ at 673 K for 2 h. The face-centered cubic (fcc) phase, named as fcc particle, was obtained by calcining the B2 particle at 673 K in air for 4 h, followed by $H_2$ reduction at 773 K for 2 h.

$N_2$ adsorption–desorption isotherms revealed that the specific surface area was 177 $m^2\,g^{-1}$ for the B2 particle while 161 $m^2\,g^{-1}$ for the fcc particle, mainly contributed by the porous silica shell generated during the high-temperature treatments under the reactive gases ($H_2/O_2$).

### STEM/ETEM analysis

STEM images were acquired over a JEOL-ARM 300 F microscope at 300 kV. Energy dispersive X-ray spectroscopy elemental mappings over the particles were collected using a JED-2300 T spectrometer. Environmental TEM observations on the particles under $H_2$ and/or $H_2$/$C_2H_2$ were done using an aberration-corrected Titan Themis ETEM G3

microscope at 300 kV. The sample was dispersed into ethanol, and the suspension was deposited onto a thermal E-chip that is equipped with a thin silicon nitride membrane. The sample was pretreated with $H_2$ at 673 K for 30 min. The images were acquired at 303 K and under 1 mbar $H_2$ or $H_2/C_2H_2$ (molar ratio of 1/1). Image simulation was done using the JEMS software package and the Multi-slice module.

## XAS spectra

XAFS spectra of Cu and Pd K-edges were measured at the BL14W1 beamline at Shanghai Synchrotron Radiation Facility, China. The sample (100–250 mg) was pressed into a self-supported wafer and mounted into a reaction cell, where it was treated with a 5.0 vol.% $H_2$/$N_2$ mixture (50 ml min$^{-1}$) at 673 K for 1 h. The spectra were then recorded at 343 K. EXAFS data were processed according to the standard procedure using Athena and Artemis modules of the IFEFFIT software packages.

## IR spectra

IR experiments were done with a dedicated ultrahigh vacuum (UHV) apparatus, combing a FTIR spectrometer (Bruker Vertex 80 v) and a multi-chamber UHV system (Prevac). 200 mg sample was pressed into an inert metal mesh and mounted on a sample holder. The sample was treated with $H_2$ at 673 K for 1 h, exposed to CO at 110 K and gradually heated to 460 K at a rate of 3 K min$^{-1}$. IR data were accumulated by recording 1024 scans with a resolution of 4 cm$^{-1}$. Peak fittings on the spectra of CO adsorbed on Pd and Cu sites were performed by the Gaussian function.

## Theoretical analysis

DFT calculations were done using the Vienna ab initio Simulation Package (VASP) code. To model the B2 and fcc particles, a 4-layer $(3 \times 2)$ B2(110) and a 4-layer $(1 \times 2)$ fcc (111) slabs were used. The bottom 2 layers were fixed in their bulk position while other metal atoms and adsorbates allowed to relax. The lattice constants of B2 and fcc phases were calculated to be 3.016 and 3.806 Å, respectively. The transition states of $H_2$ dissociation were located using the dimer method and the convergence criteria were set to 0.05 eV Å$^{-1}$. Additional computational details are reported in the Supplementary Methods.

## Catalytic tests

Selective hydrogenation of acetylene was conducted with a continuous-flow fixed-bed quartz tubular reactor (inner diameter, 6 mm) at atmospheric pressure. 50 mg catalyst (40–60 mesh) was pretreated with a 5.0 vol.% $H_2$/$N_2$ mixture (50 ml min$^{-1}$) at 673 K for 1 h. After being cooled down to room temperature (298 K), the catalyst was exposed to the reaction gas (1.0 vol.% $C_2H_2$/0.5–2.0 vol.% $H_2$/He, 50 ml min$^{-1}$) that was introduced via a mass flow controller. The outlet from the reactor was analyzed online using a gas chromatograph equipped with a thermal conductivity detector and a flame ionization detector. The reaction rate was measured by controlling the conversion of acetylene below 20% via varying the flow rate of the reaction gas or the weight of the catalyst. The activation energy was measured in the temperature range 268–318 K at $H_2/C_2H_2$ ratio of 1/1–2/1. The reaction orders with respect to hydrogen and acetylene were determined at 298 K by adjusting the concentrations of $H_2$ (0.5–3.0 vol.%) and $C_2H_2$ (1.0–3.0 vol.%) in the feed gases.

## Data availability

Source data that support the findings of this study are provided in this paper, which can also be available from the corresponding author upon reasonable request. Source data are provided with this paper.

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

## Acknowledgements

This work was supported by the National Natural Science Foundation of China (21573221, Y.L.; 21533009, W.S.; U1832174, Y.Z.; 91945302, W.L.) and the Deutsche Forschungsgemeinschaft of Germany (392178740, 426888090, Y.W.).

## Author contributions

S.L., Y.L. and W.S. prepared the catalysts and performed the reaction tests. X.Y., Y.W. and C.W. did IR experiments and data analysis. Y.Y.,

H.Z. and J.Z. conducted XAS measurements and strucutre simulations. C.Z. and W.L. did DFT calculations. S.H. and Y.Z. collected STEM/ETEM images and performed image simulations. Y.L., Y.W., W.L., C.W. and W.S. designed the experiments, discussed the data and wrote the paper. All of the authors discussed the results and commented on the paper.

## Competing interests

The authors declare no competing interests.
