## [Peer Review File · Nature Communications]

Title: Tuning crystal-phase of bimetallic single-nanoparticle for catalytic hydrogenationREVIEWER COMMENTS

Reviewer #1 (Remarks to the Author):

The authors have answered to most of my comments. However, I have a few comments below.

1- Author Reply "Moreover, previous studies have concluded that these minor facets, which we observed here on the two particles, were much less or nearly inert for the target reaction" : At least, one reference is required here.

"Cu-terminated (100) facet was inactive for acetylene hydrogenation" : At least, one reference is required here.

"Pd-terminated {100} facet catalyzed the reaction at room temperature but the consecutive Pd sites led to a much lower selectivity (Li X. et al., ACS Catal. 10 (2020) 9694–9705)." According to the reference given by the authors "we confirm that the close-packed (111)-dominated large Pdnanoparticles have indeed much better selectivity (76%) to ethene compared with the commercial Pd/C catalyst (4.5%) at 100% conversion." I do not understand the point of the authors, since the selectivity they claim (70-80%) is similar to the one of Li et al.

The arguments of the authors are questionable.

2- "An ordered L10-type (the AuCu-type) was adopted to simulate the disordered fcc (111) facet" The approximation made by the authors is now clearly stated.

The authors do not discuss the influence of the structural order. It is well known that ordered intermetallics show a increased selectivity in comparison to disordered compounds [see for example Materials 2013, 6, 2958-2977; doi:10.3390/ma6072958]. What is the relevance of using an ordered surface model while a disordered surface is identified experimentally?

3- the answer is sufficient

4- "Surface segregation for the ordered B2 particle hardly occurs, although minor change could be not fully excluded." a reference is required here.

5 to 7- the answer is sufficient

Reviewer #2 (Remarks to the Author):

Concerning the now 2nd reply to my earlier questions, I unfortunately can not see progress or additional arguments.

1) The STEM images just show, that these are not present in high concentration, still a small fraction of single atoms (likely close to the metal-silica interface) which might be very active can not be excluded by the data presented

2) A contribution of the minority phase (estimated to be ca. 10%) to the catalytic activity can not be ruled out

5) PBE is not state of the art, especially in catalysis, the revised PBE should be used as an absolute minimum. Besides, the disordered structure is calculated on ordered models. Treatment of disordered structures is possible since more than 10 years using FPLO as has been demonstrated e.g. in reference [16] on the Cu-Pd system.

6) The authors have an understanding of terms deviating from the general chemistry community: Conversion is the fraction of reactant converted in the reaction and is given in % while the activity is a material specific quantity (relating e.g. to mass, surface area, ...; unit e.g. mol(reactant converted)/(mol(transition metal)*h)) and allows the comparison of different materials. The reaction rate on the other hand is independent of a catalyst and refers to the speed of a chemical reaction, i.e. change of reactants or products over time (unit e.g. mol/h). For more details I refer to common textbooks on catalysis or physical chemistry. The terms are not used in the correct manner in throughout the manuscript.

Concerning the reported catalytic properties, it is puzzling, why a difference in activity is observed, while earlier very similar activities (after an induction period of a couple of hours) have been reported in ref [16]. In addition, ref [16] reports 20% difference in ethylene selectivity (at identical conversion levels) which was not observed in the present study. In my opinion, there is a significant influence of the silica shell on the catalytic properties (e.g. quicker diffusion of acetylene and hydrogen than ethylene through the shell) which is not taken into consideration.

Thus, I can not recommend the publication of the manuscript.

Revision Notes

Referee #1

The authors have answered to most of my comments. However, I have a few comments below.

1- Author Reply "Moreover, previous studies have concluded that these minor facets, which we observed here on the two particles, were much less or nearly inert for the target reaction": At least, one reference is required here.

Author Reply. We cited 2 references in the revised version (refs. 48 & 49, Page 12) to support this claim. These studies have demonstrated that the (100) facets of Pd or Cu were much less selective or inert for the target reaction at room temperature, and thus strongly supported our assumptions in this work.

"Cu-terminated (100) facet was inactive for acetylene hydrogenation": At least, one reference is required here.

Author Reply. As mentioned above, we cited a new reference (ref. 49) for this statement.

A recent report concluded that Cu catalysts (varying the size from single-atoms to nanoparticles) were inactive for hydrogenation of acetylene below 388 K (Shi X. et al. *ACS Catal.* 10 (2020) 3495–3504). Therefore, we could say that the Cu-terminated (100) facet was inert under our conditions, i.e., at room temperature.

"Pd-terminated {100} facet catalyzed the reaction at room temperature but the consecutive Pd sites led to a much lower selectivity (Li X. et al., *ACS Catal.* 10 (2020) 9694–9705)."According to the reference given by the authors "we confirm that the close-packed (111)-dominated large Pd nanoparticles have indeed much better selectivity (76%) to ethene compared with the commercial Pd/C catalyst (4.5%) at 100% conversion." I do not understand the point of the authors, since the selectivity they claim (70-80%) is similar to the one of Li et al. The arguments of the authors are questionable.

Author Reply. We appreciate the referee's suggestion and are sorry for this unclear statement.

The referee is correct in citing the key conclusion from the literature report, e.g., "we confirm that the close-packed (111)-dominated large Pd nanoparticles have indeed much better selectivity (76%) to ethene compared with the commercial Pd/C catalyst (4.5%) at 100% conversion." (Li X. et al., *ACS Catal.* 10 (2020) 9694–9705). Referencing the author's structure analysis, this statement implicated the much less selectivity of the commercial Pd/C catalyst (4.5%) was due to the more proportion of exposed Pd(100) facet. This result supported our view that the minor (100) facet on our PdCu particles marginally contributed the overall performance because of the less selectivity towards ethylene.

We could interpret the selectivity issue between this literature report and our work, both at room

temperature. In this report, the authors verified a higher C₂H₄ selectivity on Pd(111), compared to Pd(100), by kinetic simulations and catalytic tests. They first theoretically predicted that the energy barrier for acetylene to ethene at 298 K is 0.66 eV on Pd(111), which is much lower than that (0.91 eV) on Pd(100); while the energy barrier for over-hydrogenation of ethene to ethane is as high as 1.38 eV on Pd(111), as compared to that (0.89 eV) on Pd(100). This result evidences the preference for Pd (111) to achieve higher activity and selectivity for acetylene to ethylene. Then, their reaction tests found that Pd octahedral nanoparticles (dominated by the (111) facet) exhibited as high as 76.5% selectivity towards ethylene, but smaller Pd nanoparticles (<2 nm) had a very low selectivity of only 4.5% because of the exposure of more proportion of Pd(100). The authors compared the difference in selectivity at 100% conversion by using a very higher H₂/C₂H₂ ratio (5/1) in the reaction gas (C₂H₂/H₂/N₂ = 2/10/88), but this did not mean a high activity. From a fundamental viewpoint, this report affirms the much less contribution of Pd(100) to the selective hydrogenation of acetylene to ethylene, as we pointed out in the manuscript.

For the minor (100) facet on the B2 particle in our case, it could be terminated by either Pd or Cu atoms and these metal atoms took up only a small proportion of the total atoms exposed. Since Cu-terminated or Pd-terminated (100) surfaces are either inert or much less selective for the target reaction, we could reasonably assume that the catalytic performance was contributed by the dominantly exposed B2(110) facet.

We have clarified this point in the revised version. For accuracy, we stated as "*the minor (100) facets, terminated by either Pd or Cu in both cases, were much less selective or inactive at room temperature*" in Page 12, and cited these informative reports (refs. 48 & 49).

2- "An ordered L10-type (the AuCu-type) was adopted to simulate the disordered fcc (111) facet"
The approximation made by the authors is now clearly stated.

Author Reply. We thank the referee for this positive comment.

The authors do not discuss the influence of the structural order. It is well known that ordered intermetallics show an increased selectivity in comparison to disordered compounds [see for example *Materials* 2013, 6, 2958-2977; doi:10.3390/ma6072958]. What is the relevance of using an ordered surface model while a disordered surface is identified experimentally?

Author Reply. As the reviewer noted, the pioneering work (Friedrich M. et al., *Materials* 6 (2013) 2958–2977, i.e., ref. 16) has reported a high selectivity (> 90%) towards ethylene over the ordered PdCu particles (B2, 70 nm), under their reaction conditions (C₂H₂/H₂/C₂H₄/He = 0.5/5/50/44.5, 473 K). As generally acknowledged, the higher reaction temperature (> 393 K) or the higher fraction of H₂ could efficiently eliminate the formation of green oil (the by-product) and thus gave ethylene selectivity of 60-98% (Borodziński A. et al., *Catal. Rev.* 48 (2006) 91–144; Zhang L. et al., *Chem. Rev.*

120 (2020) 683–733). Therefore, the selectivity is intimately linked to the reaction conditions.

In our work, we found the extremely high activity of the ordered B2 particle, based on kinetic data and E-TEM experiments. We quantified the activity of the ordered B2 particle was 10 times of that for the disordered fcc particle at room temperature with a stoichiometric feed gas ($C_2H_2/H_2/He = 1/1/98$); while the selectivity towards ethylene was essentially the same (70-80%, Fig. S12), So, we focused on identifying the geometric and the electronic features of the ordered particle under our reaction conditions that are more approaching to the practical applications.

The B2 and fcc particles crystallize in the phases of ordered body-centered cubic and face-centered cubic; the coordination environment and the spatial distribution of metal (Pd/Cu) atoms vary vastly with the coordination number of 8 and 12, respectively. The ordered B2 particle could be precisely simulated by DFT calculations with a distinct Pd-Cu bond model. However, modelling the disordered fcc particle is long challenging because of the numerous possibilities for atomic arrangements. Fortunately, this disordered fcc phase has the same space packing pattern of metal atoms to the ordered L10-type structure (the AuCu-type) with the atomic coordination number of 12), as shown in the Reference Figure below. Therefore, we made a proximation by using the ordered L10-type to simulate the disordered fcc (111) facet, and our DFT calculations reasonably explained the large difference in activity. In fact, similar methods have been reported in the literature, e.g., Tong W. et al., *Angew. Chem. Int. Ed.* 132 (2020) 2671–2675 and Qiu Y. et al., *J. Am. Chem. Soc.* 140 (2018) 16580–16588). Both cases used the ordered L10-type to mimic the fcc PdCu particles and provided reasonable interpretations for the experimental observations. Following these prior works and taking our structural analysis by STEM/XAS/IR into account, we feel that our proximation by using an ordered L10-type to model the fcc phase is workable, as we discussed with the referee in the last revision.

For clarity, we detailed this point in Fig. S17 and cited relevant references.

Reference Figure. Atomic arrangements in ordered B2, disordered fcc (A1) and L10-type.

3- the answer is sufficient

Author Reply. We thank the reviewer`s positive comment.

4- "Surface segregation for the ordered B2 particle hardly occurs, although minor change could be not fully excluded." a reference is required here.

Author Reply. We deeply thank the referee for this reminding.

For the ordered B2 particle the Pd and Cu atoms are bonded *via* strong d-orbital interaction and ordered anisotropically in a specific crystallographic direction. Experimental studies have verified that they are much more stable against chemical oxidation and etching (Li, J. et al., *Acc. Chem. Res.* 52 (2019) 2015–2025; Zhou M. et al., *Chem. Rev.* 121 (2021) 736–795). Along with our IR experiments (Fig. 4 & Fig. S9), we could safely state that surface segregation for the ordered B2 particle hardly occurs, although minor change could be not fully excluded.

5 to 7- the answer is sufficient.

Author Reply. We thank the referee's help in correcting the technique errors.

Referee #2

Concerning the now 2nd reply to my earlier questions, I unfortunately can not see progress or additional arguments.

1) The STEM images just show, that these are not present in high concentration, still a small fraction of single atoms (likely close to the metal-silica interface) which might be very active can not be excluded by the data presented

Author Reply. Aberration-corrected high-angle annular dark field scanning transmission electron microscopy (HAADF-STEM) is the most powerful technique to evidence single-atoms over metal catalysts (Qiao B. et al., *Nat. Chem.* 3 (2011) 634–641; Wang A. et al., *Nat. Rev. Chem.* 2 (2018) 65–81; Hannagan R. T. et al., *Chem. Rev.* 120 (2020) 12044–12088), while Infrared (IR) spectroscopy offers additional evidences with a distinct shift in wavenumbers for CO adsorption on the metal single-atoms (McCue A. J. et al., *J. Catal.* 329 (2015) 538–546; Kruppe C. M. et al., *J. Phys. Chem. C* 121 (2017) 9361–9369; Christopher P. et al., *J. Phys. Chem. C* 122 (2018), 25143–25157; Sykes E. C. H. et al., *Chem. Rev.* 120 (2020) 12044–12088).

Recently, Pd and Cu single-atoms have been visualized by HAADF-STEM, including Cu₁-carbon nitride (Yang J. et al., *J. Am. Chem. Soc.* 143 (2021) 14530–14539), Cu₁-N-doped carbon nanotube (Jin S. et al., *Angew. Chem. Int. Ed.* 59 (2020) 21885–21889), Pd₁@SiO₂@Fe₁ (Zhao Y. et al., *Nat. Catal.* 4 (2021) 134–143), Pd₁-SiC (Chu C. et al., *Nat. Commun.* 12 (2021) 5179) and Pd₁-carbon nitride (Wang N. et al., *ACS Catal.* 12 (2022) 4156–4164). If comparing on our STEM images, both in the main text and the SI, to these reports on Pd and Cu single-atoms, we could surely exclude the presence of single-atoms. Moreover, our IR data evidenced the on-top CO-Pd band located at 2028-2048 cm⁻¹, but those for on-top CO adsorption over Pd single-atoms were observed at 2070-

2050 cm⁻¹ (McCue A. J. et al., *J. Catal.* 329 (2015), 538–546; Kruppe C. M. et al., *J. Phys. Chem. C* **121** (2017) 9361-9369). Altogether, we could conclude that the geometric arrangement of surface Pd and Cu atoms over the PdCu particles, governed by the crystal-phase, contributed to catalysis in our work.

We have highlighted these key points in the revised manuscript (Pages 5 & 10).

We highly appreciate the referee's suggestion that there might be few atoms at the metal-silica interfaces. This might be another interesting topic in our future work by specially designing such a kind of metal dispersion and quantitatively describing the catalytic behavior. By the way, single-atoms have raised a hot stream and crossed the periodic table within 10 years, but comparative studies have also revealed that the strategy of using single-atoms does not apply to all applications and heavily depends on the reaction requirements (Mitchell S. & Pérez-Ramírez J., *Nat. Commun.* 11 (2020) 4302). It is surely worthy of fundamental studies as the extreme case of metal dispersion.

2) A contribution of the minority phase (estimated to be ca. 10%) to the catalytic activity can not be ruled out

Author Reply. As we mentioned in the last revision, XRD pattern of the used B2 sample (after reaction test for 24 hours) showed minor, additional reflections that might be related to fcc phase. This is a common phenomenon for metal catalysts and generally refers to restructuring. Under the reaction conditions (temperature & atmosphere), the metal particle interacts with the molecules, approaches a new steady-state and further changes in structure with extending reaction time. This is accompanied by the variations in catalytic performance. After reaction, however, the dominate diffractions of the B2 phase kept unchanged, in line with the stable performance for 24 hours (Fig. 5a). On the other hand, even if the minor fcc phase affected catalysis, but its contribution would be marginal because the activity of the disordered fcc phase was only 1/10 of the ordered B2 phase (the topic of this work). Therefore, we conclude that the activity of the B2 particle stemmed from the ordered phase. For clarity, we have noted in Supplementary Figure 10 as "**XRD pattern of the spent B2 sample showed minor reflections that might be related to fcc phase**".

Restructuring of metal particle during the reaction course is a long-standing topic in heterogeneous catalysis; changes in particle size and shape have been intensively examined in the literature but variations in the crystal-phase, for example from ordered to disordered and vice versa, have been less explored by far. This is inspiring extensive studies recently.

5) PBE is not state of the art, especially in catalysis, the revised PBE should be used as an absolute

minimum. Besides, the disordered structure is calculated on ordered models. Treatment of disordered structures is possible since more than 10 years using FPLO as has been demonstrated e.g. in reference [16] on the Cu-Pd system.

Author Reply. The referee is correct. There do present several models for simulating ordered and disordered phases in DFT calculations. We deeply thank the referee for providing these thoughtful and invaluable inputs on this matter.

Ref. 16 has shed in-depth insights into the structure of the Pd-Cu system and its hydrogenation selectivity; the skillful DFT calculations verified the ordered PdCu had an accumulation near E_F while the disordered PdCu presented a reduction near E_F . Inspired by this interesting work, we quantified the electronic structure of the same-sized but crystal-phase-varied PdCu particles (8 nm). According to the first-round review, we have conducted DFT calculations by considering the zero-point energies on the particles and the reaction pathways. These theoretical results, along with STEM/EXAFS/E-TEM and reaction kinetics, elaborated the high activity of the ordered particle. We concluded that the Pd-Cu bond on the ordered B2 particle, featured a lower coordination number and a high-lying metal d-band center, was responsible for the extremely high activity.

Since PBE functional was proposed by J. P. Perdew, K. Burke and M. Ernzerhof in 1996, it remains one of the most popular and widely used xc functional with a total citation of more than 100,000 times, for its extensively accepted robustness and well-balanced accuracy and computational cost. As the referee noted, a revised PBE (also proposed more than 20 years ago) could improve the overbinding of reactant adsorption on surfaces in catalysis. So far, it is a formidable challenge to find a single xc-functional that is applicable for all different chemical environments. Nevertheless, theoretical studies using PBE on transition-metal catalyzed hydrogenation of alkynes have been reported recently, including PdIn (Feng Q. et al., *J. Am. Chem. Soc.* 139 (2017) 7294–7301), Pd₁Cu (Jiang L. et al., *Nat. Nanotech.* 15 (2020) 848–853), NiGa (Cao Y. et al., *Angew. Chem. Int. Ed.* 59 (2020) 11647–11652), PdAg (Li X. et al., *J. Am. Chem. Soc.* 143 (2021) 6281–6292). All these works demonstrated that PBE-based DFT calculations on acetylene hydrogenation are workable. They not only satisfactorily explained the experimental phenomenon but also reasonably predicted the general behavior of the metal catalysts.

As for the calculation on the disordered structure, ref. 16 adopted the FPLO model to treat the Cu-Pd system and provided in-depth understandings in the atomic configuration and electronic feature. Self-consistent LCAO-CPA method implemented in the FPLO code was developed for calculating electronic structure of disordered alloys, based on expanding the one-electron Green's function in the basis of modified atomic orbitals and allowing treating both the diagonal and off-diagonal disorder by using an extension of the Blackman-Esterling-Berk form of the coherent-potential

approximation (CPA) to a nonorthogonal basis set. This scheme is constrained to studies of bulk properties of disordered alloys, typically band structures and density of states, but is challenged by the complexities of heterogeneous catalysts, where properties of adsorption/desorption, diffusion and surface reaction on solid surfaces are playing key roles and exact atomic models are necessary for theoretical calculations to explore its catalytic performance.

As mentioned above by addressing the similar concern from referee 1, we made a proximation using an ordered L10-type (the AuCu-type) to simulate the disordered fcc (111) facet. We have clarified this issue in Fig. S17 and detailed our Methods in SI.

To this end, following the prior works on DFT calculations and taking our structural analysis by IR/STEM/XAS into account, we felt our computed results, by minimizing the zero-point energies as the referee suggested, are acceptable, although the accuracy needs to be further improved. We fully agree with the referee to adopt more precise models to describe the disordered crystal-phase. This would be our next step in studying other bimetallic systems.

6) The authors have an understanding of terms deviating from the general chemistry community: Conversion is the fraction of reactant converted in the reaction and is given in % while the activity is a material specific quantity (relating e.g. to mass, surface area, ...; unit e.g. mol(reactant converted)/(mol(transition metal)*h)) and allows the comparison of different materials. The reaction rate on the other hand is independent of a catalyst and refers to the speed of a chemical reaction, i.e. change of reactants or products over time (unit e.g. mol/h). For more details I refer to common textbooks on catalysis or physical chemistry. The terms are not used in the correct manner in throughout the manuscript.

Author Reply. We agree with the referee for the definitions of conversion and activity in chemical reactions. We are actually talking the same things.

It is absolutely right that the reaction rate refers to the speed of a chemical reaction, i.e. change of reactants or products over time (unit e.g. mol/h). When a catalyst is used the reaction rate may be stated on a catalyst weight ($\text{mol g}^{-1} \text{s}^{-1}$) or surface area ($\text{mol m}^{-2} \text{s}^{-1}$) basis. If the basis is a specific catalyst site that may be rigorously counted by a specified method, the rate is given in units of s^{-1} and is called a turnover frequency (TOF).

The activity is a material specific quantity and could used to compare the ability of catalysts for speeding up the reaction.

Concerning the reported catalytic properties, it is puzzling, why a difference in activity is observed, while earlier very similar activities (after an induction period of a couple of hours) have been reported in ref [16]. In addition, ref [16] reports 20% difference in ethylene selectivity (at identical

conversion levels) which was not observed in the present study. In my opinion, there is a significant influence of the silica shell on the catalytic properties (e.g. quicker diffusion of acetylene and hydrogen than ethylene through the shell) which is not taken into consideration.

Author Reply. As we mentioned above, ref. 16 reported an astonishing selectivity (> 90%) on the ordered PdCu particles (70 nm) under their reaction conditions ($C_2H_2/H_2/C_2H_4/He = 0.5/5/50/44.5$, 473 K). It could be regarded as a breakthrough in this field. For this reaction, higher reaction temperature (> 393 K) or higher fraction of H_2 in the feed stream often suppressed the formation of green oil (the byproduct) and hence gave selectivity in a range of 60-98% (Borodziński A. et al., *Catal. Rev.* 48 (2006) 91–144; Zhang L. et al., *Chem. Rev.* 120 (2020) 683–733). The differences in activity and selectivity is understandable by considering the reaction conditions. We performed the tests at room temperature (298 K) with a stoichiometric feed gas ($C_2H_2/H_2/He = 1/1/98$), and quantified the activity of the ordered B2 particle was 10 times of that on the disordered fcc particle; while their selectivities towards ethylene were very close (70-80%), due to the identical reaction mechanism (Fig. S12).

The referee is right for considering the molecular diffusion in the silica shell. However, this largely depends on the dimension of the reacting molecules and the porous structure of the silica shell. Here, we treated the samples at 673-773 K, which generated micropores on the silica shell. As illustrated by the Reference Figure below, the silica shells enriched micropores of 5.2/5.3 Å, evidenced by the high surface areas (177 m^2/g for the B2 particle; 161 m^2/g for the fcc particle). The smaller molecules, acetylene (3.3 Å) and hydrogen (2.827-2.89 Å), could diffuse onto the metal particles easily. Based on these primary experimental data, we noted in the main text (Page 4) as *"The permeable porous silica shell, formed during the high-temperature treatments with the reactive gases (H_2/O_2), spatially confined the metal particle and thus persevered the particle size, but still allowed diffusion of small molecules to the metal surface for studying their chemical transformation"*.

Reference Figure. N_2 adsorption/desorption isotherms of the PdCu particles.

Moreover, we have examined the internal and external diffusions in our kinetic studies, and the preliminary experimental data (Reference Table) excluded the impact of the silica shells.

Reference Table. External and internal diffusions on the B2 and fcc particles

Catalyst	Diffusion	Sample amount (mg)	Pellet size (mesh)	Gas flowrate (ml min ⁻¹)	Conversion (%)	Reaction rate (mol _{C₂H₂} g _{Pd} ⁻¹ s ⁻¹)
fcc particle	Internal	30.0	20–40	30	7.16	4.76×10^{-6}
		30.3	40–60	30	7.44	4.89×10^{-6}
		30.8	60–80	30	8.63	5.58×10^{-6}
	External	30.3	40–60	30	7.44	4.89×10^{-6}
		15.3	40–60	15	8.31	5.42×10^{-6}
B2 particle	Internal	10.4	20–40	70	15.86	6.74×10^{-5}
		10.0	40–60	70	14.91	6.60×10^{-5}
		10.1	60–80	70	16.57	7.27×10^{-5}
	External	10.0	40–60	70	14.91	6.60×10^{-5}
		5.0	40–60	35	14.10	6.25×10^{-5}

* The reaction tests were performed at 298 K with a feed gas of 1%_{H₂}/1%_{C₂H₂}/He.

In addition, the outweighed activity of the B2 particle has been demonstrated under other reaction conditions (Reference Figure below), which consistently supported our conclusion that the ordered phase possesses an intrinsically higher activity. We have shown these additional reaction data as Supplementary Figure 11 and emphasized this evidence in Page 10: *"The outstanding performance of the B2 particle was further demonstrated by varying the feed gas composition (Supplementary Fig. 11)."* For clarity, we have deepened our discussion on the mechanism in Page 11.

Reference Figure. Acetylene hydrogenation over the PdCu particles. **a**, conversion of acetylene; **b**, selectivity of ethylene. The reaction tests were performed at 298 K with a feed gas of 1.0 vol.% C₂H₂/1.0 vol.% C₂H₄/2.0 vol.% H₂/He (50 ml min⁻¹).

Thus, I can not recommend the publication of the manuscript.

Author Reply. We deeply thank the referee for continuously helping us to promote the quality of

this work during peer-reviews. The thoughtful and invaluable insights encouraged us to deepen fundamental understandings in the atomic structure of the particles and the reaction mechanism. We sincerely hope that these additional experimental data and our essential revisions clarified the originality of this work, especially its implications for atomic catalysis. In our personal view, the breakthroughs lie on:

- 1) We have precisely tuned the ordered and disordered phases of PdCu particles (8 nm) by coating each particle with a silica-shell, yielding the same-sized but crystal-phase-varied particles.
- 2) We have experimentally evidenced the activity of the ordered particle was 10 times greater than that of the disordered one, and correlated this astonishing performance to the densely-populated surface Pd-Cu bond with a low coordination number. This quantitatively described single-particle catalysis at the atomic accuracy.
- 3) We have directly observed, for the first time, the activation route of H₂ under the reaction conditions by E-TEM, and elaborated the reaction pathway with the aid of DFT calculations.

REVIEWERS' COMMENTS

Reviewer #1 (Remarks to the Author):

The authors have adequately addressed my comments

Revision Notes

Reviewer #1 (Remarks to the Author):

The authors have adequately addressed my comments

Author Reply. We deeply thank the reviewer`s positive comment. Again, we thank this Reviewer for the continuous and generous helps in promoting the scientific quality.